# Health Systems Determinants of Delivery and Uptake of Maternal Vaccines in Low- and Middle-Income Countries: A Qualitative Systematic Review

**DOI:** 10.3390/vaccines11040869

**Published:** 2023-04-19

**Authors:** Bronte Davies, Jill Olivier, Edina Amponsah-Dacosta

**Affiliations:** 1Health Policy and Systems Division, School of Public Health and Family Medicine, Faculty of Health Sciences, University of Cape Town, Observatory, Cape Town 7925, South Africa; jill.olivier@uct.ac.za; 2Vaccines for Africa Initiative, Institute of Infectious Diseases and Molecular Medicine, Faculty of Health Sciences, University of Cape Town, Observatory, Cape Town 7925, South Africa; edina.amponsah-dacosta@uct.ac.za

**Keywords:** vaccine, pregnancy, maternal health, health system, low- and middle-income countries

## Abstract

Maternal vaccination is considered a key component of the antenatal care package for improving maternal and child health. Low- and middle-income countries (LMICs) fall short of global targets to prevent maternal and neonatal deaths, with a disproportionate burden of vaccine-preventable diseases. Strategies towards ending preventable maternal mortality necessitate a health systems approach to adequately respond to this burden. This review explores the health systems determinants of delivery and uptake of essential maternal vaccines in LMICs. We conducted a qualitative systematic review of articles on maternal vaccination in LMICs, published between 2009 and 2023 in line with the Preferred Reporting Items for Systematic Reviews and Meta-Analysis guidelines. Thematic analysis was conducted to identify key themes in the literature, interpreted within a conceptual framing that explores the systems determinants influencing maternal vaccines. Our search yielded 1309 records, of which 54 were included, covering 34 LMICs. Most of the included studies were from South America (28/54) and included pregnant women as the primary study population (34/54). The studies explored influenza (25/54) and tetanus toxoid (20/54) vaccines predominantly. The findings suggest that systems hardware (lack of clear policy guidelines, ineffective cold-chain management, limited reporting and monitoring systems) are barriers to vaccine delivery. Systems software (healthcare provider recommendations, increased trust, higher levels of maternal education) are enablers to maternal vaccine uptake. Findings show that formulation, dissemination and communication of context-specific policies and guidelines on maternal vaccines should be a priority for decision-makers in LMICs.

## 1. Introduction

Pregnant women are more susceptible to vaccine-preventable diseases (VPDs) with more severe adverse outcomes and mortality rates compared to the general population, due to varying hormone levels, cardiopulmonary and immunologic adaptive changes to accommodate fetal growth [1,2]. Vaccination during pregnancy (‘maternal vaccination’) has been widely recognized as a key component of the comprehensive package of antenatal care (ANC) aimed at improving maternal and child health (MCH). This is because maternal vaccines not only protect pregnant women against VPDs such as tetanus, influenza and pertussis, but also protect neonates against infection or severe disease in their first weeks of life, which is oftentimes the most vulnerable period [3].

Globally, vaccinating pregnant women against tetanus has been considered a major public health success in reducing the devastating burden of the disease since the initiation of the global Maternal and Neonatal Tetanus Elimination (MNTE) program in 1989. This MNTE program reduced neonatal tetanus by 96% in targeted countries [4] and has since emerged as a platform for introduction of new and underutilized maternal vaccines [5,6]. Seasonal maternal influenza vaccination has been recommended by the World Health Organization (WHO) since 2005 [7]. After the 2009 swine flu pandemic, caused by the H1N1 influenza virus, the WHO Strategic Advisory Group of Experts on Immunization (SAGE) further recommended that pregnant women be prioritized for seasonal influenza vaccination as a high-risk group in 2012 [7,8]. Following this recommendation, wide acceptance of maternal flu vaccinations has contributed to a shift in acceptance of other maternal vaccines in many countries [8]. The diphtheria, tetanus and pertussis combination vaccine (Tdap) was first recommended for pregnant women by WHO in 2015 [9,10]. Additional vaccines recommended for administration during pregnancy in endemic countries or during outbreaks include the meningococcal A, yellow fever, cholera, Ebola, Hepatitis E, rabies and tick-borne encephalitis vaccines [3,9]. Vaccines against Group B streptococcal disease (GBS) and respiratory syncytial virus (RSV) are currently in late-stage clinical trials and have yet to be licensed for use in pregnant women [11,12]. In addition to vaccines against GBS and RSV, recent vaccines against the coronavirus disease (COVID-19) have also been targeted at pregnant women, indicating the safety and efficacy of these vaccines in preventing disease and adverse outcomes [13]. Pregnant women are classified in the high-priority use group by the SAGE Roadmap for Prioritizing use of COVID-19 vaccines, and as of early 2023, more than 200 countries and territories have policies for COVID-19 vaccination during pregnancy [14,15].

It is estimated that 89% of pregnancies worldwide occur in low- and middle- income countries (LMICs) [16]. Despite the success of global efforts such as MNTE, 18 LMICs have yet to eliminate neonatal tetanus [17]. Additionally, vaccine coverage continues to vary between countries, with some LMICs reporting coverage as low as 11% for influenza [18], while high-income countries (HICs) report coverage rates over 60% in pregnant women [19]. This is concerning considering the high burden of VPDs in LMICs, and the fact that neonatal and maternal mortality rates have declined the slowest in LMICs compared to HICs over the last decade [1,20].

Maternal vaccination efforts fall at the intersection of maternal healthcare services and routine immunization programs, usually embedded in broader health services. Despite the near-universal successes of the Expanded Program on Immunization (EPI), programmatic and systems-level challenges to routine immunization remain, leading to significant gaps in VPD prevention and control [17]. Current maternal vaccination programs in LMICs are characterized by various factors that affect delivery (on the vaccine supply side) and uptake (on the vaccine demand side) [21]. Barriers to implementation of maternal health interventions have been shown to be shared across LMIC contexts and include a limited health workforce, shortage of health resources, varied willingness of pregnant women to accept vaccines, inability to collect high-quality data due to staff shortage, inadequate funding and infrastructure, and policy implementation issues [22]. The substantial cost associated with developing new maternal vaccine delivery platforms creates an additional barrier to vaccine program implementation in LMICs where several health programs compete for limited budgets and health infrastructure [23]. Considering the shared health systems barriers in LMICs, understanding the complexities of maternal vaccinations in LMICs can guide the implementation of new and under-utilized vaccines towards strengthening MCH.

Currently, global interests aim to understand and address the barriers to effective maternal vaccination programs in LMICs. Considering the urgency to expand the reach of new and under-utilized vaccines among pregnant women, there is a need to strengthen the evidence base within this field to better inform rational policy making and implementation. The current COVID-19 pandemic and wide-scale roll-out of COVID-19 vaccines has further highlighted the need to apply a health systems lens to delivery and uptake of maternal vaccines. It is with this in mind that this qualitative systematic review was conducted to explore the question: what are the health systems determinants of maternal vaccine delivery and uptake in LMICs? The primary objectives of this review were (i) to describe the health systems determinants of the delivery and uptake of maternal vaccines in LMICs, and (ii) to improve our understanding of how these determinants may serve as barriers or enablers to establishing effective maternal vaccination programs in LMICs. Drawing on the findings of this review, policy- and practice-relevant recommendations are proposed to support current and future maternal vaccination programs in LMICs.

## 2. Materials and Methods

An exploratory systematic review study was conducted in two phases: a scoping exercise, followed by a qualitative systematic review. This review was conducted in line with the Preferred Reporting Items for Systematic Reviews and Meta-Analysis (PRISMA) guidelines [24].

### 2.1. Search Strategy

Guided by a Boolean search strategy (Appendix A) informed by an initial scoping exercise, relevant literature sources were sought through electronic databases and platforms, namely PubMed, Scopus (Medline and Embase), Web of Science, WHO Institutional Repository for Information Sharing, and EBSCOHost (Academic Premier, Africa Wide information, CINAHL, Health Source Nursing Academic, Medline, APA Psych, and APA PsycInfo). The initial literature search was conducted from October to November 2021 followed by an updated search between November 2021 and April 2023.

### 2.2. Eligibility Criteria and Study Selection

Evidence sources were only included if they met the following criteria: involved human participants in quantitative, qualitative and mixed-methods empirical studies; primary studies conducted in LMICs; published in English; and explored maternal vaccination as a subtheme or involved pregnant women as a subset of a population involved in a vaccine-related study. Searches were restricted to the literature published between 2009 until the final search date (2023). This period was identified in the preceding scoping exercise as a crucial stage in the development of research within the field of maternal vaccination in LMICs and includes key landmarks such as the recommendation of Tdap, influenza and COVID-19 vaccines for pregnant women [9]. Studies were deemed ineligible if they met the following criteria: did not directly measure or explore maternal vaccination; measured only epidemiological outcomes in relation to vaccine products; were solely conducted in HICs. The literature from search yields were exported into Zotero (v.61) reference manager, and then imported into the Rayyan systematic review management online platform where deduplication and screening of titles, abstracts and full texts were performed [25].

### 2.3. Study Appraisal

The quality of empirical quantitative, qualitative and mixed-methods studies was appraised during review of relevant full-text records by BD and EAD, using the Joanna Briggs Institute Critical Appraisal Checklists and the Mixed Methods Appraisal Tool [26,27]. Studies considered to be ‘low quality’ by these checklists were not automatically excluded from the research, particularly if findings were relevant, unless the study design and/or results were considered unreliable and there was minimal congruity between research philosophy, aims and methods.

### 2.4. Data Extraction

The first author (BD) extracted data from the included studies guided by a data extraction sheet designed for this review and informed by the preliminary scoping exercise. The core variables sought for extraction included findings on decision-making, financing, implementation and delivery, the health workforce and information systems as they pertained to maternal vaccination programs in LMICs. A summary of the data extracted from the studies included in this review is provided in Appendix A.

### 2.5. Conceptual Framework

Exploring health systems elements that contribute to maternal health outcomes, in the context of vaccination programs, requires an understanding of the interaction among the delivery of maternal healthcare services, vaccination programs and health systems. Various conceptual frameworks for health systems research exist and have evolved over time, ranging from descriptive and analytical to deterministic and predictive [28]. Shared health systems critical functions, otherwise classically known as ‘building blocks’, include: service delivery, health workforce, health information, technologies and commodities, demand generation, financing and governance [28,29].

A framework that describes the ‘health systems shelter’ which has been applied for exploration of MCH has also been proposed previously, and offers a useful basis for unpacking systems complexities in terms of health systems hardware, software and values [30]. While systems hardware refers to tangible systems building blocks such as resources and financing, software refers to people, processes, networks and behaviors that drive systems [31]. Complementing this framework is a previously published logic model that integrates health systems critical functions with national immunization program components of: (i) policy, standards, and guidelines, (ii) governance, organization, and management, (iii) human resources for health, (iv) vaccine cold-chain and logistics, (v) service delivery, (vi) communication and community partnerships, (vii) data generation, and (viii) sustainable financing [17,32]. Logical frameworks specifically addressing analysis of immunization services in the context of health systems have been applied to LMIC contexts [33]. Considering the burden of VPDs and maternal mortality in LMICs, the WHO’s Strategies Toward Ending Preventable Maternal Mortality points to a systems approach to MCH in terms of hard- and software [34].

Drawing on these previous frameworks, a conceptual framing was applied during this systematic review to guide interpretation of the findings when describing, analyzing and making sense of how health systems hardware, software and contexts interact to influence delivery and uptake of maternal vaccines. The description of health systems determinants of delivery and uptake as hard- and software allows for an improved understanding of how and why determinants interact to produce the maternal vaccination outcomes observed in LMICs [29]. The capacity of health systems to support maternal vaccination programs in LMICs is not comprehensively captured in the published literature, necessitating the development of a new conceptual framing to address this gap for the purpose of this systematic review. In this framing, health systems are interpreted by the context in which they exist, indicating how delivery and uptake of maternal vaccines falls at the intersection of vaccination programs and maternal healthcare. Furthermore, this framing incorporates elements of the ‘health systems shelter’ framework developed by Agyepong et al. [30] and the logic framework for assessing the interaction between health systems and national immunization programs developed by Amponsah-Dacosta et al. [30]. For the purpose for this review, ‘health systems determinants’ refers to any factors relating directly to health systems, including but not limited to the system building blocks of service delivery, human resources, information systems, medical products, financing and/or leadership and governance [29].

### 2.6. Data Synthesis

Key findings from the qualitative studies helped contextualize quantitative data through identification of concepts using thematic analysis [35]. Thematic inductive analysis was applied by BD to develop themes and to elicit rich meaning from the extracted data, as has been carried out in previous reviews on vaccine uptake and delivery [36,37,38]. Themes were described through complex interpretation of the extracted data, to reveal or identify underlying complexities at the intersection of health systems and maternal vaccination programs. Building on the descriptive themes, analytical themes allowed application of the conceptual framing to inform the evidence synthesis for this review. This conceptual framing enables the abovementioned descriptive themes of delivery and uptake to be interpreted and understood through the barriers and enablers of systems hardware and software, with careful consideration for the prevailing context in which the studies were conducted.

## 3. Results

The electronic database search identified 1309 records. Once duplicates were removed and titles and abstracts were screened for eligibility, 95 relevant records were identified. Upon full-text review and appraisal, 41 records were excluded because these studies reported outcomes that were not related to maternal vaccines in general (*n* = 35), failed to meet appraisal requirements due to low methodological quality (*n* = 1), or were not conducted in LMICs (*n* = 5). A total of 54 articles were judged to be eligible for inclusion in the final systematic review (Figure 1). All studies included were published between 2012 and 2023. 

### 3.1. Characteristics of Included Studies and Participants

All records included in this review were published, peer-reviewed articles, the majority (36/54) of which were qualitative or quantitative cross-sectional studies. Of the remaining records, six adopted a mixed-methods study design, one was a randomized control trial, and one a cohort study. Overall, the quality of included studies was moderate to high, although articles which used qualitative designs tended to omit acknowledgement of authors’ reflexivity and positionality within the study (Appendix A).

The distribution of the literature by country and vaccine of focus is presented in Figure 2. In total, 34 LMICs were represented in the included literature, with studies from Kenya, Tanzania, South Africa, Malawi, Ivory Coast, Sierra Leone, Zambia, Senegal, The Gambia, Nigeria, Burkina Faso, Rwanda, Mozambique, Guinea, India, Argentina, Mexico, Brazil, Peru, Honduras, Chile, Paraguay, Uruguay, El Salvador, Nicaragua, China, Thailand, Taiwan, Bangladesh, Iran, Indonesia, Malaysia, Vietnam and Morocco. Regionally, studies from South American countries predominated the evidence base on maternal vaccination (25/54), with a notable gap in evidence from Central and North Africa, Eastern Europe and the Eastern Mediterranean regions.

With regards to the types of vaccines studied, the literature appears to be dominated by investigations focused on tetanus toxoid (20/54) and influenza (25/54) vaccines administered during pregnancy (Figure 2). The remainder of studies focused on the Zika virus vaccine (3/54); the tetanus, diphtheria and pertussis (Tdap) vaccine (2/54); the COVID-19 vaccine (3/54); the pertussis vaccine (1/54) and maternal vaccines in general without specific focus on any one vaccine type (5/54). Several studies investigated administration of more than a single vaccine type.

The following population groups participated in the studies included in this review: pregnant women (34/54); community members (17/54); healthcare providers ranging from community health workers to obstetricians (13/54); and policymakers (4/54) (Table 1). Special population groups such as migrants, or high-risk groups such as pregnant women or women of child-bearing age living with HIV/AIDS, were not represented. Where healthcare providers were concerned, a heavy focus on public sector providers was noted, except for a small proportion of studies which explored providers’ influence on maternal vaccine delivery and uptake in the private health sector (8/54). Also worth noting was the absence of studies that explored the role of other key health actors, such as community leaders, traditional healers, faith-based actors and non-profit/non-governmental providers in maternal vaccination programs.

### 3.2. Health Systems Determinants of Maternal Vaccine Delivery

For the purpose of this review, ‘maternal vaccine delivery’ refers to supply-side factors that affect optimal provision of vaccines. Policy, leadership and governance emerged as one of the determinants of delivery of maternal vaccines in LMICs (Table 2). Of the four studies that explored experiences of policymakers and health leadership across multiple LMICs, two found that the majority of countries studied had a maternal vaccine policy or guideline in place [39,40]. In Kenya [41], Argentina, Brazil, Honduras, Mexico and Peru [40], the formation and dissemination for such policies and guidelines were experienced as being predominantly of a top-down approach, as described by policy- and decision-makers. This approach to policy formulation and dissemination served as a barrier to effective delivery of maternal vaccines, resulting in negative effects as seen in Thailand [42] China [39,43] and South Africa [44] where healthcare providers reported experiencing poor communication on appropriate maternal vaccine guidelines from their national governments to facility-level staff. At the facility level, the lack of clearly communicated policies and guidelines on maternal vaccination resulted in critical vaccine implementation gaps. Here, implementation refers to vaccine distribution, cold-chain management, coverage, utilization and availability of platforms for delivery of maternal vaccines. In Brazil [40] and Kenya [41], adoption of a top-down approach to policy formulation on maternal vaccination excluded consultation with relevant stakeholders such as primary care clinicians and community-based organizations. Reliance on guidelines from the WHO and Centers for Disease Control and Prevention in place of context-specific national guidelines was reported in Taiwan, highlighting the lack of a coordinated, contextually appropriate approach to vaccination policy development [45]. The experience in Argentina, Brazil, Honduras, Mexico and Peru was similar, in that the lack of co-ordination was evident in the discrepancies in vaccine delivery guidelines for healthcare providers working in the public, private and informal health sectors [46]. An in-depth study of Kenya’s National Immunization Technical Advisory Group (NITAG) revealed that their maternal vaccination policy borrowed from vaccine policies of other countries [41], which could be categorized as a systems hardware barrier to maternal vaccine delivery.

Further hardware determinants were described in terms of availability of sustainable financing mechanisms to support maternal vaccination programs. In South America (Honduras, Brazil, Mexico, Peru and Argentina), the use of pooled procurement funding mechanisms earmarked for implementing maternal vaccine programs was identified as a health systems enabler of effective vaccine delivery [40]. Most countries in this region were reported to rely on domestic funding, with the exception of Honduras which receives supplementary funding for procurement of maternal vaccines from Gavi, the Vaccine Alliance [40]. While Thailand reported health insurance as being a contributor to funding ANC [42], a multi-country study indicated that the majority of LMICs rely more on patient out-of-pocket payments than health insurance for ANC including maternal vaccination services (Table 2) [47]. Funding sources were also reported to influence vaccine decision-making and formulation of recommendations needed to guide the delivery of maternal vaccines in LMICs. In Kenya, decision-makers reported that reliance on international and donor funding was a key enabler as the availability of funding supported the feasibility of recommendations [41]. On the other hand, a lack of dedicated budgets at the facility level was identified as a barrier to providing educational resource materials on vaccination to healthcare providers and pregnant women in Kenya [46].

Shortage and unavailability of maternal vaccines was another prominent theme in the evidence base (Table 2). In most LMICs, procurement of vaccines was predominantly through the EPI, with the exception of Honduras, Brazil, Mexico and Peru where vaccines were procured through ANC or MCH services [39]. Stockouts of vaccines emerged as a systems hardware barrier to delivering vaccines to pregnant women in South Africa [18], Kenya [48] and India [49] (Table 2). Compounding this situation further were reports of the intricate cold-chain management and storage requirements which facilities in some LMICs found difficult to maintain, for various reasons. In Kenya, for example, improper vaccine cold-chain resource management was quoted as being a result of their national devolution process [48], whereas in Tanzania cold-chain limitations were the direct result of power cuts affected by seasonal weather patterns and delayed procurement of fuels such as gas [50]. Furthermore, in Kenya, limited availability of transport for vaccines from county depots to facilities with proper cold storage mechanisms was a barrier to effective vaccine delivery [51]. This meant that despite willingness to be vaccinated, pregnant women reported the limited availability of vaccines to be an issue, impacting on the quality of care received, which was a barrier also described in South Africa [44]. The manner in which certain maternal vaccination programs were implemented also posed additional challenges, as seen in the South American region where seasonal influenza campaigns are restricted to certain months of the year, limiting delivery and subsequent uptake [40,46,52,53].

Eight studies conducted in Kenya [48], India [49,52,54], Thailand [42], Brazil [55], Ethiopia [56], and El Salvador [57] explored maternal vaccine delivery and uptake in the private health sector. Overall, these studies underscored the existence of weak co-ordination between public and private health sectors, as well as national immunization and ANC and/or MCH programs where maternal vaccine delivery is concerned. In both public and private health contexts, delivery of maternal vaccines varied across platforms and facilities, with maternal vaccines being delivered at primary health clinics, antenatal clinics and hospitals. Two main platforms for delivery of maternal vaccines were reported in the literature, namely the EPI and ANC. The frequency of attendance to ANC varied by country and context. A key enabler to higher vaccine coverage, in the case of tetanus and influenza vaccines, was found to be increased frequency of antenatal visits in LMICs such as Sierra Leone, Ethiopia and India (Table 2) [47,58,59,60]. This positive association was reported across countries, regardless of vaccination timing during pregnancy, although countries such as Kenya and Ecuador reported delivery of vaccines to pregnant women for the first time during second and third trimesters [59,61].

Experiences of healthcare providers, ranging from facility nurses to hospital-based obstetricians, further highlight systems barriers to vaccine delivery during pregnancy. Staff shortages and a lack of skilled human resources for health emerged as a prominent theme in this regards, augmented by heavy workloads at facilities and, in some cases, high staff turnover [48,62]. In Kenya, delivery of vaccines to pregnant women was hindered by health worker strikes [63]. Health worker shortages and heavy workloads were found to contribute to long waiting times for pregnant women to receive vaccines, serving as barriers to vaccine delivery in India, Mexico and Brazil [50,64]. Healthcare providers in Nicaragua and India described initiatives such as vaccine administration to women in waiting lines at antenatal clinics as an attempt to overcome such barriers and facilitate adequate coverage [52,65]. Adequate knowledge about VPDs and vaccine effectiveness demonstrated by healthcare providers was strongly associated with sound recommendations to pregnant women, across multiple contexts [42,54,66]. In countries where healthcare providers fueled vaccine hesitancy or rather promoted non-pharmaceutical interventions for VPDs, knowledge on the diseases and vaccine safety was generally reported to be low [42,67]. This was particularly highlighted amongst physicians, who are somewhat neglected in terms of training and capacitation around maternal vaccination, as nurses are generally the focus of continuing professional education programs [18,39,46]. While willingness of healthcare providers to recommend maternal vaccines was low in China, increased willingness to recommend these vaccines was associated with younger age of healthcare professionals and higher professional title [68]. Skepticism around maternal vaccines and unaddressed concerns about their safety and efficacy among healthcare providers was a key theme across findings emerging from China [68]. It has been suggested that this lack of vaccine endorsement among healthcare providers stems from irregular updates to national guidelines or a lack of local guidelines altogether. Limited access to clear policies and guidelines on maternal vaccination compromises healthcare providers’ roles as vaccines advocates and delivery agents [43,68,69]. Gaps in healthcare provider communication in turn negatively impact service delivery and compromise adequate vaccine coverage among pregnant women. Where official policies were not available, providers in Kenya and China reported relying on regular updates on maternal vaccines through national bulletins, and, in some cases, peers and colleagues to support them in executing their role [45,49].

Another key health systems determinant of maternal vaccination delivery in LMICs is the health information and reporting system which countries employ. In this review, this theme was mainly explored at the facility level, where information and reporting systems relied on vaccination cards and ANC booklets [59,63]. Lack of accurate reporting of vaccine coverage has been attributed to the reliance on physical vaccination cards or ANC booklets that are frequently misplaced, contributing to challenges in monitoring of subsequent vaccine doses administered (as is required for the Tdap and tetanus vaccines) [59,70]. In the absence of any records, this also meant that healthcare providers often relied on patient recall of prior doses administered [59]. Monitoring processes and systems were explored in two studies conducted in Malawi and several South American countries [18,64]. Notably, in both studies, there was a lack of formal reporting structures for Adverse Events Following Immunization (AEFI) [18,64]. A multi-country investigation into vaccine safety data tools identified a lack of technical and human resources at the facility level to be a barrier to effective surveillance and monitoring [71]. In studies that explored delivery by both public and private health sectors, it was highlighted that information sharing between private and public facilities was non-existent, subsequently revealing a lack of co-ordination between sectors as a barrier to sustainable maternal vaccination efforts [40,48,52,57]. This relates specifically to a persistent lack of strong information systems to monitor vaccine rollout and coverage amongst pregnant women utilizing services across both health sectors. Ultimately, the reliance on systems hardware such as physical ANC records and vaccination cards, in place of coordinated electronic tracking systems, serves as a barrier to effective service delivery.

Finally, it is important to address the role of local contextual determinants on the delivery of vaccines to pregnant women in LMICs. Contextual influences on maternal vaccine delivery identified in the studies included in this review highlight strong political will as an enabler to maternal vaccine policy implementation. This was described as a key factor in maternal vaccine decision-making by health leadership in Brazil [40]. In Kenya, healthcare providers reported that the local political landscape did not interfere with vaccine delivery at the facility level [51]. Notably, gangsterism and violent crime served as a prominent contextual barrier to procurement and delivery of maternal vaccines in El Salvador. This is because fear of violence limits pregnant women’s access to facilities, where they require permission to access healthcare or where they avoid facilities entirely (Table 2) [57].

### 3.3. Health Systems Determinants of Maternal Vaccine Uptake

For the purposes of this review, uptake refers to the receipt, acceptance and utilization of vaccination services by a pregnant women and women of childbearing age. Factors influencing individual decision-making processes are a key determinant in whether pregnant women will accept lifesaving vaccines. In all 54 studies included in this review, the role of healthcare providers, spouses, relatives and community members as major influencers for pregnant women in terms of their willingness to accept vaccines was addressed [43,49,53,64,67,69,72,73]. The most influential factor identified was the recommendation of healthcare providers to pregnant women, which leverages the elements of trust in the health system that almost all women and community members expressed across the literature [57,64,74,75,76,77]. Healthcare provider recommendations were reported to be the driving factor for vaccine uptake during pregnancy in LMICs such as Ecuador [61] Mexico [78] and Malaysia [79], as well as in Peru [80], where maternal knowledge of vaccines during pregnancy was found to be low and uptake was reliant on provider recommendations (Table 3). In Nicaragua, maternal socio-demographic factors such as age, ethnicity and employment status were reportedly not associated with vaccine uptake but rather uptake was entirely dependent on recommendations by healthcare providers [81,82]. In Kenya, recommendations came solely from ANC healthcare providers [51]. Trust in the healthcare system and healthcare provider recommendations can thus be considered to serve as systems software enablers of maternal vaccine uptake (Figure 3). This theme on trust as an enabler to uptake extended to how pregnant women perceived their national and local governments. For example, in Kenya, pregnant women expressed that they trusted the government to only recommend vaccines that are safe and effective [77]. In other LMICs, however, unstable political contexts tended to jeopardize this level of trust, and women were hesitant to accept vaccines in Guinea [83] or viewed vaccine policies as control mechanisms motivated by political and/or financial gain, as was demonstrated with the maternal influenza vaccination campaign in Morocco during the H1N1 pandemic (Table 3) [75].

In the eight studies that explored pregnant women’s access to maternal vaccination in Kenya [63,76], Brazil [52,71,84], El Salvador [57], Ethiopia [56] and Uganda [73], it was evident that disparities in access existed between services provided by the public versus the private health sectors. The public sector was consistently reported to administer vaccines during pregnancy free-of-charge. However, not all countries offer free antenatal services, and thus services provided during an antenatal visit might require out-of-pocket payment even if the vaccine administered during that same visit was free, as was observed in Thailand [39,42]. Also notable were the experiences in the private healthcare sector, where pregnant women tended to access ANC through the private sector but would then be referred to a public facility just to receive their vaccines because of the free access [54]. Notably, user fees for these services were shown to be associated with a lower vaccine coverage for tetanus and influenza vaccines [45,58,85,86]. In Malawi, an authoritative and punitive approach to vaccine policy implementation where pregnant women incur penalties if ANC is missed, in an effort to incentivize ANC attendance and increase vaccine coverage, rather led to unintended outcomes including low vaccine uptake among pregnant women (Table 3) [64].

While healthcare providers are a source of trusted information and education about VPDs, it was frequently reported that constraints to service delivery (such as staff shortages and high workloads) in turn affect uptake. This is because healthcare providers with high workloads tend to have limited time to spend with each patient for health education and thorough communication about maternal vaccines, as reported in Kenya [72] and Thailand [62], indicating that human resource constraints serve as a hardware barrier to vaccine uptake during pregnancy, thus highlighting the interplay between health systems determinants of vaccine delivery and uptake as shown in Figure 3. Additionally, pregnant women and community members’ previous negative experiences with health service delivery—although not necessarily related to vaccination—serve as a software barrier to uptake of maternal vaccines in LMICs. In Ethiopia and Brazil, a commonly cited reason for this was mistreatment and disrespect from healthcare providers, deterring pregnant women from antenatal visits and subsequent vaccine uptake [52,56]. This is concerning given the reliance on healthcare providers in individual decision-making about vaccine uptake. The theme of awareness and education identified in the literature highlights the need for credible information sharing and seeking by relevant population groups. Women in Brazil reported reliance on healthcare providers for vaccine information, but also on certain media sources which have been shown to peddle misinformation about vaccine safety. Consequently, reliance on such media sources creates barriers to vaccine uptake and fuels mistrust in healthcare providers’ recommendations [71].

Cultural and social norms and beliefs varied by context and interestingly tended to influence reasons for accessing private healthcare and individual willingness to pay for maternal vaccination services. These contextual systems software factors were evident in beliefs held by pregnant women in Kenya who perceived that paying for services incentivizes healthcare providers to share sufficient information on vaccines and VPDs [63], while in Malawi paying for health services was associated with good social standing as a status symbol, particularly among male spouses who deem it honorable to access paid services (Table 3) [57]. In Morocco, however, women preferred accessing public services for vaccination during pregnancy for the reason that free vaccine provision was not profitable for the facility and thus incentives were altruistic and reasons for vaccination were rooted in effectiveness and not profit [68]. Culture and religion were also found to be a barrier to vaccine uptake in some LMICs. This was the case in a study conducted in Senegal where a sub-population of pregnant women belonging to a specific religious group were barred by group members from accepting vaccines during pregnancy [67].

Sociodemographic factors such as maternal education, race, socioeconomic status, and parity appeared to have a heterogenous effect on decisions to accept a vaccine during pregnancy, depending on the setting studied. Reports on the effect of contextual variables such as level of maternal education, access to transport and place of residency (urban versus rural) varied across the included studies. In South American countries, Kenya and Ethiopia , increased vaccine coverage was associated with urban residency, as pregnant women in rural areas reported extreme distances to ANC facilities [48,52,86,87], whereas in Indonesia [88], rural residency was associated with increased coverage and uptake. In Kenya, a higher level of education among pregnant women was associated with low acceptance of vaccination services [59]. In Ethiopia, contextual factors that affected uptake included higher level of maternal education, having a TV in the house, occupational status, place of birth, and general knowledge of available vaccines [56,89,90,91,92]. In the same country, receiving more than two or more recommended doses of the tetanus vaccine during pregnancy was associated with urban residency, short travel distance to health facilities and higher level of maternal education (at least secondary school) [90]. A higher level of maternal education was also associated with increased vaccine acceptance of the COVID-19 vaccine in Vietnam [91]. In Ivory Coast, increased vaccine uptake was seen among pregnant women with higher education, younger age, rural residency and attendance of at least three ANC visits [92]. Notably, however, other studies conducted in Iran, Kenya, The Gambia and Senegal found no significant association between vaccine coverage and acceptance and these same sociodemographic factors (residency and level of maternal education) [45,58,75,93]. When it came to maternal occupation, it was notable that in El Salvador there was mention of the role of employers where pregnant employees were not granted time off work to access ANC during work hours [57], serving as a contextual barrier to uptake of vaccines.

## 4. Discussion

This systematic review reports synthesized evidence on the health systems determinants of delivery and uptake of maternal vaccines in LMICs. The dynamic interaction between key barriers and enablers of delivery and uptake, categorized into health systems hardware and software, are also described and provide unique insights into the resultant suboptimal performance of maternal vaccination programs observed in some LMICs. If we are to tackle the global Sustainable Development Goal to promote maternal health beyond just survival [84], then it is pertinent to implement context-specific public health interventions. Such interventions should be guided by robust evidence on the gaps in delivery and uptake of healthcare services including vaccination. Policy- and decision-makers will rely on such evidence in order to develop feasible recommendations that are immediately important for scaling-up the performance of health systems with consequent improvements in health outcomes.

### 4.1. Findings and Recommendations for Vaccine Decision- and Policy-Makers

The identified barriers to effective delivery and uptake of maternal vaccines in LMICs which could be classified as systems hardware include the lack of coordinated, context-specific approaches to vaccine policy formulation, reliance on public funding mechanisms and out-of-pocket payments, poor cold chain systems and management processes, weak health workforce capacity including staff shortages and limited capabilities, and reliance on paper-based vaccine monitoring and reporting systems [42,54,67]. It was observed that, where clear policies and guidelines on maternal vaccination were in place, this supported vaccine delivery and subsequently promoted uptake among pregnant women [39,40]. It is important to address the fact that despite maternal vaccination falling at the intersection of ANC and immunization programs, lack of service co-ordination was identified as a prominent barrier to maternal vaccine delivery and uptake in LMICs [42,45,76,80]. This lack of co-ordination was demonstrated between public and private health sectors, and between EPI and ANC services. What this suggests is a gross under-utilization of the opportunities to leverage existing infrastructure and delivery models to improve maternal health outcomes, at least where reducing the devastating burden of VPDs is concerned [94]. Service co-ordination remains a priority strengthening area that could reduce service delivery cost by leveraging existing infrastructure to deliver vaccines. This would address identified barriers to vaccine uptake, which include the cost of education materials and the reliance on donor funding for vaccination programs at a local level [23,95]. Such efforts could also mitigate maternal out-of-pocket payments for accessing key healthcare services [39]. Establishing coordinated service delivery of currently available maternal vaccines is paramount to sustainable introduction of new and upcoming maternal vaccines such as those against COVID-19, GBS and RSV [11,12,13]. Sustainable introduction of new maternal vaccines could in turn present unique opportunities for health systems strengthening through coordinated health service provision. This echoes findings in other LMIC settings, where the introduction of new vaccines into immunization programs has been shown to improve co-ordination between ministries of health and other government ministries, such as education and social development [96]. The resource and infrastructural barriers to vaccine delivery in LMICs, demonstrated by vaccine stock-outs and ineffective cold-chain maintenance processes [18,52,61], have been reported previously [8]. Often, this is exacerbated by system shocks such as the recent COVID-19 pandemic, which affect access to services in LMICs. A survey exploring the impact of COVID-19 on immunization programs reported that 53% of respondents in LMICs had experienced disruption to delivery of maternal vaccines, almost 10% more than in HICs [97]. These system shocks, although on different scales, require sustainable heath system strengthening approaches. The fact that limited healthcare provider capacity was identified as a barrier to maternal vaccine delivery in LMICs [48,62] underscores the need for a health systems strengthening response which is designed to ensure that new maternal vaccine introductions are coupled with increasing the number of staff at facilities and providing regular staff training and appropriate compensation or incentives in accordance with potential short-term increases in workloads brought on by the introduction of additional or new maternal vaccines [96]. Furthermore, the issue of healthcare provider capacity should be addressed through coordinated delivery of vaccines between EPI and ANC, as well as co-ordination and partnerships between public and private healthcare providers [23]. Finally, weak health information and reporting systems characterized by the reliance on paper-based vaccine monitoring and reporting systems and maternal recall require particular attention in order to support the establishment of effective and sustainable maternal vaccination programs in LMICs [64]. The reliance on maternal recall for immunization history is not limited to LMICs and has also been observed previously in HICs [98,99]. To this end, strengthening health information systems through reliable VPD surveillance and monitoring of maternal vaccine coverage would enhance policy- and decision-making, while strengthening research and patient centered care, as well as minimizing the reliance on HIC data [39]. The availability of AEFI reporting systems in only two of the LMICs studied in this review points to the lack of robust monitoring systems, and over-reliance on clinical trial data and passive surveillance in these settings [100]. Incorporation of active surveillance and monitoring of AEFIs is urgently required and could be coupled with the introduction of new vaccines, as has been implemented previously in HICs [95].

In terms of health system software, trust in political governance and healthcare provider recommendations on maternal vaccines emerged as a prominent theme in the evidence base on maternal vaccine delivery and uptake. There was a particular emphasis on effective communication channels between healthcare workers and clients of the health system, including pregnant women and their close contacts who play an important role in influencing their individual decisions on vaccine acceptance and uptake [74,80,85]. Having access to reliable sources of information on maternal vaccines could counter the misinformation and disinformation peddled by some media forums and religious and cultural organizations who fuel mistrust in vaccines. Evidently, there is a need to improve communication between pregnant women and healthcare providers, in order to promote vaccine uptake. This requires service delivery platforms to steer away from ‘unidirectional’ communication between healthcare providers and pregnant women, and rather encourage broader community engagement throughout the vaccination program cascade (from the development to the implementation of vaccines). The use of tools such as antenatal records and vaccination cards to enhance communication and consultation with the broader community is also worth exploring [95]. Such activities would be far-reaching in also promoting trust between healthcare providers, pregnant women and the broader community served by the health system. This is not restricted to LMICs, however, as promotion of trust in healthcare providers, as well as the importance of vaccine efficacy and safety communication to pregnant women, have also been identified as enablers to uptake and delivery in HIC settings [98,99]. Mistrust in vaccines and vaccine hesitancy is also not limited to LMICs but has been qualitatively recorded in HICs [99]. This calls for improved education and communication strategies that address the concerns expressed by pregnant women and improve their knowledge and awareness about maternal vaccination. Extensive knowledge of maternal vaccination appears to be associated with higher income status [98]. In this review, the influence of low level of education among women, including broader community members, was highlighted as a key sociodemographic factor and barrier to vaccine uptake [56,91,92], demonstrating that a health systems strengthening approach to vaccine programming in general requires thinking beyond just the health sector. Rather, initiatives should extend strengthening efforts to improving education, socioeconomic conditions and financial protection [100,101].

Another important health systems’ software barrier to the delivery of maternal vaccines in LMICs is the ineffective dissemination of policies and guidelines from the national level to healthcare providers at the facility level. Globally, across both HICs and LMICs, it has been shown that healthcare provider awareness of vaccination policies enables them to recommend influenza vaccination during pregnancy, second to vaccination of the healthcare providers themselves [102]. This emphasizes the need for improving communication of policy from health leadership to healthcare providers at the facility level [39,44,68]. A clear dissemination process that also includes facility-level staff in policy- and decision-making processes would bridge the gap for some vaccine implementation issues and promote ownership and accountability among healthcare providers. This could further improve trust between healthcare providers and pregnant women, given a shared vested interest in the vaccination program, thereby facilitating increased uptake.

As health systems continue to adopt maternal vaccination policies for new and future vaccines such as for COVID-19, GBS and RSV, careful consideration of system-wide effects is paramount. The findings of this review have significant implications for establishing maternal vaccination programs which effectively deliver existing as well as new and future vaccines. Such efforts will require a strengthened delivery response relying on the WHO’s recommendation of at least eight ANC visits during pregnancy [103]. Future vaccine introduction would also benefit from effective formulation and communication of policies and guidelines which are appropriately disseminated to the facility level to sensitize healthcare providers to critical practices such as gestation administration windows. The recent introduction of new vaccines against diseases such as COVID-19 is an opportunity for increasing acceptance of maternal vaccines by spouses, family and community members, who have been shown to influence willingness to vaccinate and subsequent uptake of vaccines by pregnant women in multiple settings [104].

### 4.2. Findings and Recommendations for Health Systems and Maternal Vaccination Research Agenda

The exploratory nature of this review allowed for identification of several gaps in the health systems and maternal vaccination research agenda for LMICs. In countries where a high level of trust in healthcare providers was reported [57,64,74,75,76], uptake of maternal vaccines was observed to be high. The promotion and maintenance of trust in healthcare providers could also be paramount to clinical trials of new maternal vaccines, which rely on the trust between research teams and community members [104]. In conducting research amongst pregnant women, particularly for clinical trials of future vaccines, maintaining and promoting trust among researchers, community members and healthcare providers is crucial for systems strengthening and has been shown to contribute to successful trial outcomes [104].

We identified four main areas for improvement of the research agenda relating to maternal vaccination and health systems, building on the evidence gaps identified in this review. Firstly, there is a need for further exploration of interventions for health promotion and education during pregnancy with a particular focus on preventing VPDs. This has been investigated for HICs [99], but not adequately in LMICs. Secondly, there is a need to better understand the role of policymakers such as NITAGs and how they influence maternal vaccine delivery and uptake in LMICs. Only one study conducted in Kenya explicitly reported on the involvement of NITAGs in maternal vaccination programs [41]. Their role in maternal vaccination should be further explored, particularly with regards to evidence-based policy formulation and clear messaging regarding vaccine safety and efficacy. Thirdly, there is a need to ensure that community trust and key stakeholder engagement are foundational pillars of maternal vaccination programs. Strategies in this regard should be the focus of future research in the field. Lastly, further research on vaccine implementation for pregnant women residing in LMICs within Central Africa, North Africa, Eastern Europe and Eastern Mediterranean regions is needed to build the knowledge base and support evidence-informed interventions where needed. Ultimately, our findings point to the need for innovative complex health-systems-based approaches to strengthening delivery and uptake of maternal vaccines in LMICs. Such approaches should be explored in future feasibility studies.

### 4.3. Strengths and Limitations of this Review

A total of 54 published sources, adopting various study designs and involving diverse stakeholders (pregnant women, community members, healthcare providers, policymakers and health leadership) from across 34 LMICs were triangulated and assessed with guidance from a conceptual framing model developed in a preceding scoping exercise. To our knowledge, this is the first ever qualitative systematic review on the health systems determinants of maternal vaccination in LMICs. Given the influence of variabilities in context on the core outcome measures assessed in this review, it is important to caution that findings may not be generalizable to all LMICs. The evidence in this review stems predominantly from descriptive and cross-sectional studies that utilize questionnaires and surveys for data collection. Such methodology could result in over-representation of certain population groups based on response bias, limiting generalizability to all LMIC health systems. Additionally, cross-sectional enquiries describe current health system determinants of vaccine delivery and uptake at the time of study and do not explore trends in vaccine uptake/delivery over time. It is also notable that the articles included in this review had an over-representation of pregnant women with minor representation of some key stakeholders such as policy- and decision-makers. Finally, while a robust search strategy was used to retrieve the relevant literature, it is possible that other publications were missed, possibly as a result of indexing within databases and restriction to English records.

## 5. Conclusions

This qualitative systematic review contributes towards improving our narrow understanding of the health systems determinants of the performance of maternal vaccination programs in LMICs. Considering the systems’ software and hardware determinants of maternal vaccine uptake and delivery identified in this review, it is evident that formulation, dissemination and communication of context-specific policies and guidelines on maternal vaccines should be a priority focus for decision-makers in LMICs. In addition, there is a need to strengthen the role of healthcare providers as change-agents and champions of maternal vaccination. Improving knowledge and awareness of VPDs, as well as the safety and efficacy of vaccines, among pregnant women and their close contacts who influence their individual decision-making should be prioritized. Governments in LMICs are encouraged to strengthen co-ordination between ANC, national immunization programs, and private and public health sectors in order to increase ANC visits and vaccine uptake during pregnancy. As new and improved maternal vaccines are introduced to the market, it is recommended that health systems determinants are taken into consideration during program implementation processes so as not to exacerbate existing barriers to maternal vaccine uptake and delivery, with the ultimate goal of establishing sustainable maternal vaccination programs.

## Figures and Tables

**Figure 1 vaccines-11-00869-f001:**
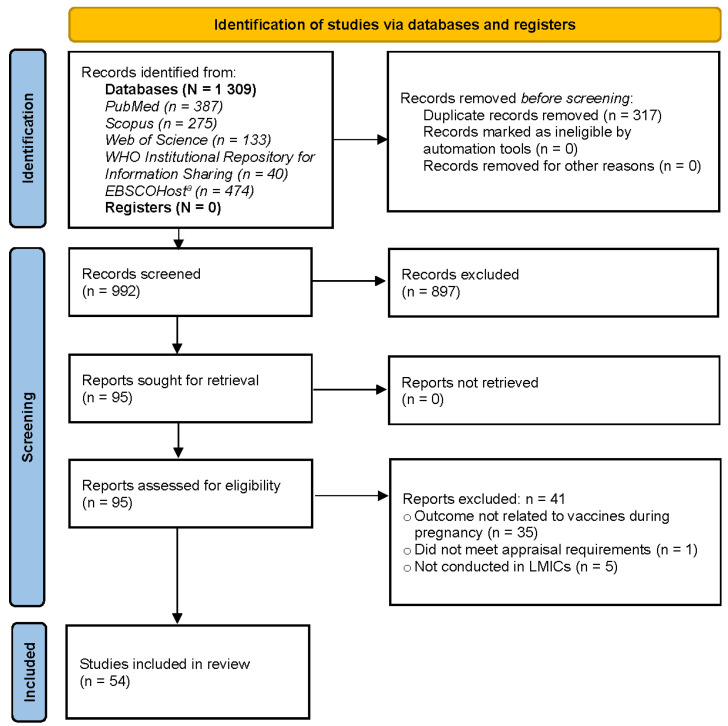
PRISMA flow diagram. Identification, screening and inclusion of literature for this qualitative systematic review [24]. ^a^ Databases searched within EBSCOHost included Academic Premier, Africa Wide information, CINAHL, Health Source Nursing Academic, Medline, APA Psych, and APA PsycInfo.

**Figure 2 vaccines-11-00869-f002:**
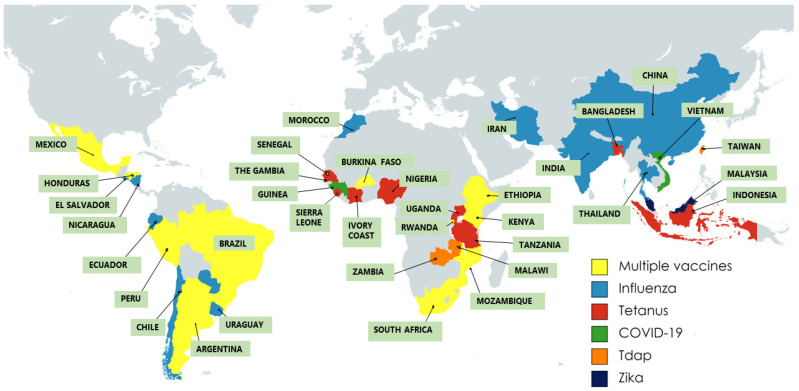
Geographical distribution of the included studies, by vaccine type. Multiple vaccines refer to studies investigating administration of more than one maternal vaccine type. This figure refers only to the maternal vaccines reported in included studies and does not reflect the entirety of vaccines available in the countries represented here. Tdap: tetanus, diphtheria, acellular pertussis.

**Figure 3 vaccines-11-00869-f003:**
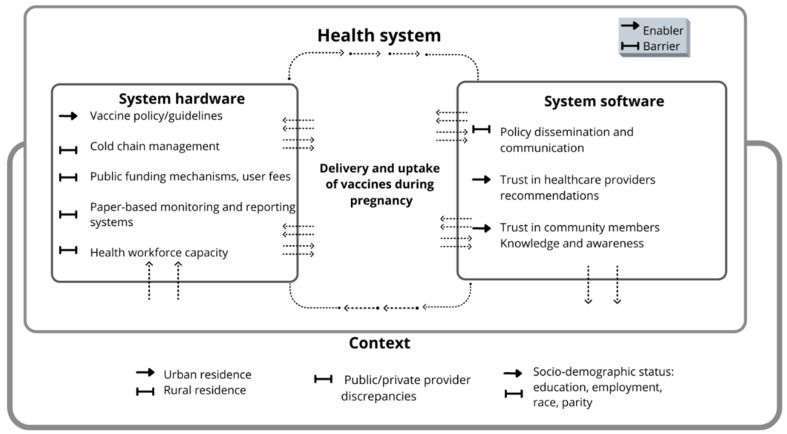
Conceptual framing of health systems determinants of delivery and uptake of maternal vaccines in LMICs.

**Table 1 vaccines-11-00869-t001:** Representation of participants involved in included studies on maternal vaccination in LMICs.

Population Group	Description	Total Studies (*n* = 54)	Total Participants (*n*)	
Pregnant women	Women pregnant at time of study	34	80,814	
Community members	Women of childbearing age, partners of pregnant women, family, broader community	17	85,566	
Healthcare providers	Obstetricians, physicians, nurses, community health workers	13	1326	
Policymakers and health leadership	National technical advisory group members, health ministry, facility managers, researchers	4	153	

**Table 2 vaccines-11-00869-t002:** Descriptive themes on health systems determinants of maternal vaccine delivery in LMICs.

Health Systems Determinant	Descriptive Themes Relating to Maternal Vaccine Delivery (Supply-Side)	
	Enablers of Vaccine Delivery	Barriers to Vaccine Delivery
Policy, leadership and governance		Top-down policy formulation and implementation omits key stakeholder engagement (Brazil, Kenya, Thailand). Lack of coordinated national approach to vaccine policy formulation leads to reliance on international guidelines without consideration for local contexts (Taiwan, Argentina, Brazil, Honduras, Mexico, Peru). Lack of communication from leadership to facility-based healthcare providers on maternal vaccination guidelines (China, Thailand, Kenya).
Financing	Pooled procurement funding for vaccine delivery services (Brazil, Mexico, Honduras, Peru). Reliance on donor funding informs feasibility of vaccine policy recommendations at national level (Kenya).	Reliance on out-of-pocket payments negatively affects pregnant women’s access to lifesaving vaccines. Low facility-level budgets compromise vaccine education and awareness efforts (Kenya).
Implementation: vaccine products and service delivery	Increased ANC visits are associated with higher coverage of influenza and tetanus vaccines among pregnant women (Sierra Leone, Ethiopia, India)/	Guideline discrepancies for vaccine administration and timing during pregnancy between public and private providers (Brazil, Mexico, Honduras, Peru). Vaccine stock-outs (South Africa, Kenya, Brazil). Lack of fuel, electrical power and transport lead to vaccine shortages and impact cold-chain management (Taiwan, Kenya).
Health workforce	Training of nurses supports maternal vaccine delivery (Honduras, Brazil, Peru).	Staff shortages due to strikes or high turnover (Kenya, El Salvador). Heavy staff workloads (Nicaragua, India). Lack of clinician training and professional development (Honduras, Brazil, Peru).
Information systems	Reliance on ANC booklets/vaccine cards and paper-based systems (Burkina Faso, Mozambique, Kenya, Sierra Leone).	Lack of formal monitoring and reporting structures (Malawi, South Africa). Lack of public/private provider co-ordination (Thailand, Kenya). Lack of electronic health records and/or effective integration thereof (Uganda).
Context	Strong political will for maternal vaccines (Brazil). Lack of political interference at the facility level (Kenya).	Gangsterism and crime limit pregnant women’s access to facilities (El Salvador).

ANC: antenatal care.

**Table 3 vaccines-11-00869-t003:** Descriptive themes on health systems determinants of uptake of maternal vaccines in LMICs.

Health Systems Determinant	Descriptive Themes Relating to Maternal Vaccine Delivery (Demand-Side)	
	Enablers of Vaccine Uptake	Barriers to Vaccine Uptake
Policy, leadership and governance	High level of trust in political governance (Kenya, Ethiopia).	Low level of trust in political governance (Morocco).
Decision-making	Healthcare provider recommendations. High level of trust in healthcare providers.	Media sources fuel mistrust (Brazil).
Financing	Out-of-pocket payments associated with social standing by spouses (Malawi).	Out-of-pocket payments. User fees for ANC services.
Implementation: access		Punitive approach to missed ANC appointments (Malawi).
Health workforce	Education and awareness of maternal vaccines provided by healthcare providers.	Low level of trust in healthcare providers due to previous negative experience (Ethiopia, Brazil).
Information systems		Reliance on maternal recall in place of patient records.
Context	Higher education level of pregnant women. Rural residency (Indonesia). Urban residency (Kenya, Bangladesh).	Higher education level of pregnant women (Kenya). Rural residence. Increased travel distance to facilities. Belonging to a religious group (Ethiopia, Senegal). Maternal employment—employers do not grant time off work to access ANC, including vaccination (El Salvador).

ANC: antenatal care.

## Data Availability

Search strategy and a summary of extracted data is provided in Appendix A. Any additional data will be provided by the corresponding author upon reasonable request. Search strategy and a summary of extracted data is provided in Appendix A. Any additional data will be provided by the corresponding author upon reasonable request.

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
