# Peer review of "Health Systems Determinants of Delivery and Uptake of Maternal Vaccines in Low- and Middle-Income Countries: A Qualitative Systematic Review"

_vaccines, 2023, doi:10.3390/vaccines11040869_

Round 1

Reviewer 1 Report

This is a well written piece which is consistent with the Special topic edition.

Several minor points.

Line 35 explain why this is the case, namely, pregnancy involves a change in the immune state leading to ........

COVID-19 vaccines, has there been any revised advice since the arrival of the Omicron variant?

Literature search conducted in late 2021. Is there need for a quick refresh?

Vaccine delivery guidelines: do you have any examples of good and poor guidelines? What were hthe key issues in a 'poor' guideline.

Would a complex systems approach to resolving problems be appropriate? Several publications by WHO on this topic.

Final very minor point: do a global replace of 2 spaces with 1 space.

This review is a useful contribution to the topic.

Author Response

Point 1: Line 35 explain why this is the case, namely, pregnancy involves a change in the immune state leading to ........

Response 1: We thank the Reviewer for this comment. For better clarity, lines 35-37 have been accordingly revised as: “Pregnant women are more susceptible to vaccine-preventable diseases (VPDs) with more severe adverse outcomes and mortality rates compared to the general population, due to varying hormone levels, cardiopulmonary and immunologic adaptive changes to accommodate foetal growth [1, 2].”

Point 2: COVID-19 vaccines, has there been any revised advice since the arrival of the Omicron variant?

Response 2: Yes, there have been developments with regards to COVID-19 vaccination during pregnancy. Lines 65-67 have been updated accordingly to include reference to country policy tracker for COVID vaccines during pregnancy. The revised sentence now reads, “Pregnant women are classified in the high-priority use group by the SAGE Roadmap for Prioritizing use of COVID-19 vaccines, and as at early 2023, more than 200 countries and territories have policies for COVID-19 vaccination during pregnancy [14,15].”

Point 3: Literature search conducted in late 2021. Is there need for a quick refresh?

Response 3: We agree with the Reviewer that a refresher search is required to update this systematic review and ensure that we synthesize and report on a comprehensive body of evidence. We have now conducted an updated search covering the period from 2021 up to April 2023. This search yielded 7 relevant articles that have now been included in the revised manuscript. Relevant sections of the manuscript have now been updated to reflect this, including the methods section in lines 119-120 which now reads as follows: “The initial literature search was conducted from October to November 2021 followed by an updated search between November 2021 and April 2023.”.

Point 4: Vaccine delivery guidelines: do you have any examples of good and poor guidelines? What were the key issues in a 'poor' guideline.

Response 4: The Reviewer’s comment is well noted. We argue that for this review, the purpose was to assess and aggregate primary research that may be informed by guidelines on maternal vaccination, rather than review guidelines directly. The challenges identified with such guidelines were not necessarily in terms of the content of the guidelines themselves but in the implementation of the guidelines. This includes a lack of clarity and specificity for vaccine dose and timing; lacking contextually appropriate information within available guidelines, and guidelines at local level contrasting those at national/international levels. This is outlined in lines 268-287 of the manuscript which reads as follows:

" Of the four studies that explored experiences of policy-makers and health leadership across multiple LMICs, two found that the majority of countries studied had a maternal vaccine policy or guideline in place [39,40]. In Kenya [41], Argentina, Brazil, Honduras, Mexico and Peru [40], the formation and dissemination for such policies and guidelines were experienced as being predominantly of a top-down approach, as described by policy- and decision-makers. This approach to policy formulation and dissemination served as a barrier to effective delivery of maternal vaccines, resulting in negative effects as seen in Thailand [42] China [39,43] and South Africa [44] where healthcare providers reported experiencing poor communication on appropriate maternal vaccine guidelines from their national governments to facility-level staff. At facility level, the lack of clearly communicated policies and guidelines on maternal vaccination resulted in critical vaccine implementation gaps. Here, implementation refers to vaccine distribution, cold-chain management, coverage, utilisation and availability of platforms for delivery of maternal vaccines. In Brazil [40] and Kenya [41], adoption of a top-down approach to policy formulation on maternal vaccination excluded consultation with relevant stakeholders such as primary care clinicians and community-based organizations. Reliance on guidelines from the WHO and Centers for Disease Control and Prevention in place of context-specific national guidelines was reported in Taiwan, highlighting the lack of a coordinated, contextually-appropriate approach to vaccination policy development [45]. The experience in Argentina, Brazil, Honduras, Mexico and Peru was similar, in that the lack of co-ordination was evident in the discrepancies in vaccine delivery guidelines for healthcare providers working in the public, private and informal health sectors [46].”

Point 5: Would a complex systems approach to resolving problems be appropriate? Several publications by WHO on this topic.

Response 5: The Reviewer raises a critical point with regards to carefully considering the complexity of health systems and that of maternal immunization programmes when designing interventions to improve the delivery and uptake of maternal vaccines. There is a need for locally generated evidence obtained through implementation research in order to better inform such innovative and feasible complex systems approaches. Given the importance of these approaches, we have raised this as part of our recommendations in lines 675-678 which now reads as follows: “Ultimately, our findings point to the need for innovative complex health systems-based approaches to strengthening delivery and uptake of maternal vaccines in LMICs. Such approaches should be explored in future feasibility studies.”

Point 6: Final very minor point: do a global replace of 2 spaces with 1 space.

Response 6: We thank the Reviewer for identifying this typographical error. All spelling, grammatical and formatting errors have now been corrected.

Reviewer 2 Report

Thank you for the opportunity of reviewing the paper “Health systems determinants of delivery and uptake of maternal vaccines in low- and middle-income countries: a qualitative systematic review” by Davies and coll. Overall, the research is well written and fits the journal’s general audience.

Please, find here my comments and suggestions.

Abstract: concise and well structured

Introduction: Offers a complete view of the problem

Methods:

- The fact that the first part of the work is part of first author’s Master of Public Health research project can be remove from the main text. If authors want to maintain this reference, it could be reported in acknowledgement section.

- Line 124-126: how is the time window defined? It may be useful to add some references.

Results:

Line 204, Please, avoid starting with a number. Possible re-phrasing could be “The electronic database search identified 1,242 records.”

Line 221, “acceptable” should be changed with a more appropriate term. Use term of Table S3

Please, be sure that all acronyms used in Tables and Figures (for example, Table 3) are explained in Table’s description.

Discussion:

Limitation section is poor. Authors might want to consider the principal limitation of the included reports and how these could impact findings and conclusions of this SR

Author Response

Point 1: Methods: The fact that the first part of the work is part of first author’s Master of Public Health research project can be remove from the main text. If authors want to maintain this reference, it could be reported in acknowledgement section.

Response 1: The Reviewer’s comment is well noted. Line 108 been revised to, “An exploratory systematic review study was conducted in two phases: a scoping exercise, followed by a qualitative systematic review.” The statement relating to the Master of Public Health research project has now been moved to the Author Contributions section (Lines 721-729).

Point 2: Methods:  Line 124-126: how is the time window defined? It may be useful to add some references

Response 2: We thank the Reviewer for flagging this. This search period was identified during the scoping review as integral to the development of the field of maternal vaccination. Line 129-130 has been revised accordingly: “This period was identified in the preceding scoping exercise as a crucial stage in the development of research within the field of maternal vaccination in LMICs and includes key landmarks such as the recommendation of Tdap, influenza and COVID-19 vaccines for pregnant women..” This scoping review can be found at https://open.uct.ac.za/handle/11427/37140.  

Point 3: Results: Line 204, Please, avoid starting with a number. Possible re-phrasing could be “The electronic database search identified 1,242 records.

Response 3 : As suggested by the Reviewer, line 209 has been rephrased to:The electronic database search identified 1,309 records.”

Point 4: Results: Line 221, “acceptable” should be changed with a more appropriate term. Use term of Table S3

Response 4:  The Reviewer’s comment is well noted. In line 228,  “acceptable” has been replaced with “moderate to high” as recommended.

Point 5: Please, be sure that all acronyms used in Tables and Figures (for example, Table 3) are explained in Table’s description.

Response 5: Table 2 and Table 3 legends have been revised to include explanations for the acronym ANC (antenatal care).

Point 6: Limitation section is poor. Authors might want to consider the principal limitation of the included reports and how these could impact findings and conclusions of this SR

Response 6: We thank the Reviewer for this comment. The Strengths and limitations section (Lines 679-697) has been revised as follows:

“A total of 54 published sources, adopting various study designs and involving diverse stakeholders (pregnant women, community members, healthcare providers, policymakers and health leadership) from across 28 LMICs were triangulated and assessed with guidance from a conceptual framing model developed in a preceding scoping exercise. To our knowledge, this is the first ever qualitative systematic review on the health systems determinants of maternal vaccination in LMICs. Given the influence of variabilities in context on the core outcome measures assessed in this review, it is important to caution that findings may not be generalizable to all LMICs. The evidence in this review stems predominantly from descriptive and cross-sectional studies that utilise questionnaires and surveys for data collection. Such methodology could result in over-representation of certain population groups based on response bias, limiting generalisability to all LMIC health systems. Additionally, cross-sectional enquiries describe current health system determinants of vaccine delivery and uptake at the time of study and do not explore trends in vaccine uptake/delivery over time.It is also notable that the articles included in this review had an over-representation of pregnant women with minor representation of some key stakeholders like policy- and decision-makers.  Finally, while a robust search strategy was used to retrieve relevant literature, it is possible that other publications were missed, possibly as a result of indexing within databases and restriction to English records.”